# 'It is not fashionable to suffer nowadays': Community motivations to repeatedly participate in outreach HIV testing indicate UHC potential in Tanzania

Josien de Klerk[1]☯, Arianna Bortolani[2]‡, Judith Meta[1]‡, Tusajigwe Erio[1], Tobias Rinke de Wit[1]‡, Eileen Moyer[1]☯*

1 Amsterdam Institute for Global Health and Development, Amsterdam, the Netherlands, 2 Doctors with Africa–CUAMM, Padova, Italy

☯ These authors contributed equally to this work.
‡ These authors also contributed equally to this work.
* e.m.moyer@uva.nl

**Data Availability Statement:** Data cannot be shared publicly because of confidentiality issues and agreements made with research participants.

## Abstract

### Objective

This study examined people's motivations for (repeatedly) utilizing HIV testing services during community-based testing events in urban and rural Shinyanga, Tanzania and potential implications for Universal Health Coverage (UHC).

### Methods

As part of a broader multidisciplinary study on the implementation of a HIV Test and Treat model in Shinyanga Region, Tanzania, this ethnographic study focused on community-based testing campaigns organised by the implementing partner. Between April 2018 and December 2019, we conducted structured observations (24), short questionnaires (42) and in-depth interviews with HIV-positive (23) and HIV-negative clients (8). Observations focused on motivations for (re-)testing, and the counselling and testing process. Thematic analysis based on inductive and deductive coding was completed using NVivo software.

### Results

Regular HIV testing was encouraged by counsellors. Most participants in testing campaigns were HIV-negative; 51.1% had tested more than once over their lifetimes. Testing campaigns provided an accessible way to learn one's HIV status. Motivations for repeat testing included: monitoring personal health to achieve (temporary) reassurance, having low levels of trust toward sexual partners, feeling at risk, seeking proof of (ill)-health, and acting responsibly. Repeat testers also associated testing with a desire to start treatment early to preserve a healthy-looking body, should they prove HIV positive.

Data are available from the Amsterdam Institute for Global Health and Development Institutional Data Access (contact via secretariat@aighd) for researchers who meet the criteria for access to confidential data.

**Funding:** The Shinyanga and Simiyu Test&Treat program in Tanzania is supported by Gilead Sciences (USA) and the Diocese of Shinyanga through the Good Samaritan Foundation (Vatican). The implementation of the project is by Doctors with Africa CUAMM and the Diocese of Shinyanga within the framework of the Tanzanian Ministry of Health, Community Development, Gender, Elderly and Children (MoHCDGEC) through the National AIDS Control Program (NACP). The scientific evaluation of the project is under guidance of Principal Investigator Prof Anton Pozniak and is performed by the Amsterdam Institute for Global Health and Development (AIGHD) in collaboration with CUAMM. This research falls under the overall research project 'Feasibility of universal access to HIV test and treat in Shinyanga and Simiyu Regions, Tanzania, ethically cleared by NIMR under NIMR/HQ/R.8a/Vol.IX/2711'. The content of this manuscript is solely the responsibility of the authors and does not necessarily represent the official views of any of the institutions mentioned above. We thank all our institutional collaborators, the study participants, the staff at the project clinical sites and laboratories, as well as the project support staff for their invaluable support to this program in general and the current manuscript in particular.

**Competing interests:** The Shinyanga and Simiyu Test & Treat program in Tanzania is funded by Gilead Sciences (USA), contract date June 1st 2016. The Amsterdam Institute for Global Health and Development (AIGHD) is a co-recipient of this award. All authors were employed under AIGHD's auspices. This does not alter our adherence to PLOS ONE policies on sharing data and materials. The funders did not play an active role in study design, data collection, analysis, decision to publish or preparation of the manuscript. All authors declare not to have competing interests.

## Conclusions

Community-based testing campaigns serve three valuable functions related to HIV prevention and treatment: 1) enable community members to check their HIV status regularly as part of a personalized prevention strategy that reinforces responsible behaviour; 2) identify recently sero-converted clients who would not otherwise be targeted; and 3) engage community with general prevention and care messaging and services. This model could be expanded to include routine management of other (chronic) diseases and provide an entry for scaling up UHC.

## Introduction

In 2016, Treat All, referring to the immediate initiation of all people living with HIV on antiretroviral treatment, was implemented as the standard of HIV care in Tanzania [1] in alignment with the 2011 UNAIDS 90-90-90 targets, which aim to identify 90% of all HIV-infected people, link and retain 90% of HIV-positive people to antiretroviral treatment (ART), and achieve viral suppression among 90% of those on treatment [2]. The strategy in Tanzania is called Treat All, referring to the immediate initiation of all people living with HIV on antiretroviral treatment. In this context, a pilot UTT intervention combined with decentralized HIV care was designed and simultaneously rolled-out in the Tanzanian Shinyanga and Simiyu regions. This project is hosted by the Catholic Diocese of Shinyanga, supported by an international donor, implemented by an Italian organisation and evaluated by a Dutch Global Health research institute. The intervention informs the Tanzanian government about the most effective UTT practices. This paper describes people's motivations for (repeatedly) utilizing HIV testing services during community-based outreach events in urban and rural Shinyanga.

Between May 2017 and April 2018, 91,968 [3] HIV tests were administered, through community-based testing campaigns and during special testing events in rural and urban locations in Simiyu and Shinyanga Regions, including Shinyanga town, a regional centre and hub for truck routes. Campaigns specifically targeted groups that were usually underserved by health facility-based testing, such as youth and men [4, 5].

Because of repeat testing practices, no data were available on how the total number of tests performed corresponded to the total number of clients. Initial results showed that the outreach reached more men (56%) than women and 31.9% of respondents were under 20 years [3]. In April 2019 the percentage HIV positive cases newly identified in the campaigns was 1.8%, lower than the reported 2016 HIV prevalence for these regions (5.9% for Shinyanga and 3.9% for Simiyu) [6]. Tanzanian testing coverage by August 2019 was estimated to be 78% [7]. With this already high coverage rate's the significantly lower percentage of newly identified HIV patients was not surprising. Declining 'yields', percentages of HIV positive patients identified through community out-reach campaigns, have motivated African governments [8] and international donors [9] (to embrace targeted HIV testing approaches directed at those considered most-at-risk. Tanzania is in a transition in this respect with a recent move to expand index testing and targeted provider-initiated testing in health facilities [10]. Qualitative data collected from the preliminary phase of our research, conducted between May 2017 and April 2018, uncovered some unexpected findings. At least half (51,1%) of the population was repeat HIV testers (Giulia Martelli, personal communication) and of those who did test HIV positive, 13% had already been enrolled in HIV services and received ART [3]. These data raised several

questions about the phenomenon of repeat testing. With so many people testing HIV negative, so many having already tested at least once previously, and a sizeable percentage already enrolled in AIDS treatment programmes, what motivated different people to utilize community-based testing campaigns and what could this imply for future programming and community outreach?

In this paper we present the findings of a qualitative ethnographic study to understand motivations for utilizing HIV testing services, including repeat testing, during community-based testing events in urban and rural Shinyanga, Tanzania.

## Methods

### Ethical clearance

Research for the project was conducted with the approval of the Tanzanian National Institute for Medical Research. The research falls under the overall research project 'Feasibility of universal access to HIV test and treat in Shinyanga and Simiyu Regions, Tanzania,' ethically cleared by NIMR under NIMR/HQ/R.8a/Vol.IX/2711.

### Study setting and project background

Shinyanga Region, Tanzania, is mainly rural and home to diamond and gold mines, which attract migrant workers. The region has two urban centres: the regional capital of Shinyanga Town and the larger mining town of Kahama. Both are hubs for businesses and transport. The population of Shinyanga Region was 1,534,808 in 2012, with a male:female ration of 96:100, that of Simiyu Region was 1.584,157, with a male:female ratio of 92:100 [11]. Most residents belong to the Sukuma ethnic group and live in scattered homesteads; both women and men travel for work to other regions, including for seasonal farming. Government dispensaries, health clinics, and private faith-based health centres offer HIV testing, as does the Shinyanga regional hospital and a host of private laboratories. HIV outreach testing also takes place during important events on public holidays. Shinyanga has 8 hospitals, 23 health centres, 215 dispensaries and Simiyu 8 hospitals, 18 health centres and 192 dispensaries (Shinyanga and Simiyu Regional AIDS Control Coordinators, personal communication).

The Shinyanga and Simiyu Test and Treat (T&T) project, within which this ethnographic study was conducted, is implemented by CUAMM-Doctors with Africa in collaboration with the Catholic Diocese of Shinyanga which runs the four 'hubs'–Care and Treatment Centres (CTC)– selected to identify HIV-positive people through provider-initiated testing and counselling as well as mobile outreach to communities in remote areas. The original aim was to test 300,000 people in Shinyanga and Simiyu regions to identify an estimated 20,000 HIV patients and link them to care. Distances from the campaign sites to these hubs varied between 0.5 and 5 hours cycling. We used cycling hours as this was the most common form of transportation to the clinic and was used by clinic counsellors as well. Mobile outreach follows the 'kitongoji strategy', offering services at the ward level to make testing accessible. To date, the T&T project has targeted the wards of 172 villages. One hub is located in an urban location and outreach activities take place at busy crossroads, bus-stands, marketplaces and densely populated neighbourhoods. In rural areas, seasonal and daily agriculture rhythms dictate people's availability for testing; in town, work and school hours affect who can participate in testing campaigns, hence testing events were held during the day and in the evenings and on public holidays.

The feasibility and effectiveness of the intervention is being studied by a multidisciplinary team, based at the Amsterdam Institute for Global Health and Development. This paper draws on data collected between May 2018 and April 2019 within a broader ethnographic study on people's experiences with Treat All and differentiated care. Our applied study entailed an

inductive, bottom-up approach, with attention to themes that emerged from rounds of data-collection and built-in opportunities in the protocol to adjust our data collection tools. Methodologically, participant observation and informal conversations informed the development of the in-depth interview tools.

Study findings were used to inform the implementation on a continuous basis. Interim results were discussed with the implementing organisation, CUAMM, in the quarterly testing team meetings. Regional sharing took place through the Quarterly Regional HIV/AIDS Partners meeting at the Shinyanga and Simyu Regional Medical Office. The primary beneficiaries of the programme were the implementing organisation, including the counsellors running the testing programme, secondary beneficiaries included the district and regional health authorities, and government officials.

## Data collection

Two Tanzanian Master's level cultural sociologists (one male, one female) were trained in ethnographic methods before conducting observations of the community-outreach campaigns in two catchment areas, an urban and a rural site. One researcher was a native Sukuma speaker. The sociologists were trained in structured observation, note-taking, probing, transcription and data-analysis, and they worked with a detailed observation guideline. Research focused on community mobilization techniques, social dynamics in the waiting area, and practices in counselling sessions. When people expressed their motivation for testing during counselling sessions, researchers recorded it. To recruit people undergoing testing, rapid interviews were held with people in the waiting area. Each person was asked to provide basic demographic information, contact details, and consent to be (re-)interviewed one month after the test. Following analysis of the rapid interviews, a diverse sample was purposively selected based on gender, age, geographical location, testing history, and HIV status. These people were then approached by telephone to schedule an in-depth interview (IDI), to which 8 HIV-negative and 2 HIV positive people responded positively. To increase the sample of HIV-positive participants, people who tested positive were specifically approached through the implementing partner. Here a convenience approach was used since no relationship with researcher had been established.

Interview guides for IDI were translated from English into Kiswahili. IDIs lasted 60 to 90 minutes. Before the start of the interview each respondent was taken through a written informed consent procedure and given the opportunity to withdraw from the study. Interviews were conducted in Kiswahili. Although this is the national language, some people living in the area spoke only rudimentary Kiswahili and these were interviewed in Kisukuma by a Sukuma-speaking researcher. Informed consent of illiterate participants was obtained by an extended verbal explanation of the study and elicited verbal consent. The respondents chose the setting for the interview, in locations ranging from private homes to public cafés to workplaces. Interviewees were asked to recount their testing experience, their motivation for testing, the testing process and trajectory, and their views on the new Treat All strategy. Example questions included: 'Can you tell me about the day you tested, what convinced you to go to get tested? How did the counsellor do the test? (probe for the testing practice). What do you know about Treat All? (probe for health benefits and prevention benefits). All tools were translated into Kiswahili and pretested with volunteers to check for flow and comprehensibility. Because recruitment was excessively difficult, we limited pre-testing to a few participants per tool (both men and women).

Additionally, four focus group discussions were conducted with influential leaders from the community in the areas surrounding the urban and the rural health centre. These were chosen

### Data analysis

Analysis took a two-step approach. Rapid interviews were initially analysed for testing characteristics of community-based testing campaign attendees. In addition to basic demographics, codes included 'number of previous tests and location of tests', 'reason to come for testing at the campaign'. Based on these interim findings, IDI guides were developed. All IDI were audio-recorded, and transcribed in Kiswahili by a trained data-transcriber and checked against the tape by the social scientists. They were all translated into English for the benefit of the wider study, but the social scientists analysed the Kiswahili version. IDI were analysed together with the observations and content of the rapid interviews. Content analysis was conducted using NVivo 12 Pro (QSR international, Doncaster, VC, Australia), since we were interested in the unexpected themes emerging from the data. The goal of our analysis was to understanding the meaning of themes, i.e., repeat testing, for different community groups, health professionals and the implementing organisation. A set of codes was developed by the first author (JK) and the Tanzanian social scientists (JM, TE) through an interactive process: Preliminary codes were based on the initial research objectives, subsequent codes were developed through a line by line reading of interviews were read line by line and a discussion of the meaning by the group, and a code was created. Main coding categories followed the trajectory of most clients: "Testing Experiences", including sub-codes such as testing trajectory, motivation for testing, re-testing, testing refusing, testing location, alternative treatment history, reaction HIV+ result. 'Counselling' included sub-codes such as client concerns, 'script'/language, counseling HIV+, counselling HIV-, reaction to HIV+ result, counselling practice, sero-discordancy.

The two Tanzanian social scientists reflected on their own positionality in weekly meetings with the lead social scientist. These included conversations about gender, language, age and how these informed rapport as well as conversations on the ethics of getting involved. In the analysis, several biases were acknowledged, including the sampling strategy for HIV-positive testers. Rather than selecting equal numbers of males and females, youth or older people, we sampled on categories of testing outcome: linked to care, not linked to care. Recruitment proved more difficult and time consuming than expected, especially amongst those not linked to care. The consequences of this is visible in the final sample of clients living with HIV, which in the rural area comprises more men than women. We addressed this bias by carefully analysing gender dynamics in testing decisions and decisions to link or not link to care after an HIV-positive test. Another bias constituted age. IDIs with HIV-negative clients of testing campaigns consisted of many older men and relatively few young women. We address this bias in age by looking at the historical time in which generations grew up and situating their answers in these time periods.

## Results

### Study participant characteristics

The total number of interviewees was 72, including people who were approached for a rapid interview (41, 2 HIV-positive) at testing campaigns and the people interviewed through an IDI (31, 23 HIV-positive). The interview data were complemented with 24 observations of outreach campaigns. Table 1 presents an overview of the testing characteristics of the participants of the rapid interviews.

**Table 1. Testing characteristics participants rapid interviews (n = 41).**

|  | Ngokolo (Urban) | | | Bugisi (Rural) | | |
|---|---|---|---|---|---|---|
|  | Male | Female | Total | Male | Female | Total |
| First time to test | 2 | 1 | 4 | 1 | 2 | 3 |
| Re-tested 1–3 times | 4 | 7 | 11 | 0 | 5 | 5 |
| Re-tested 4–9 times | 4 | 3 | 7 | 5 | 3 | 8 |
| Re-tested 10+ times | 2 | 1 | 3 | 0 | 0 | 0 |
| Unknown | 1 | 0 | 1 | 0 | 0 | 0 |
| Total | 13 | 12 | 25 | 6 | 10 | 16 |

Amongst the HIV-negative clients of the CBTC (41), 8 were first-time testers, most of whom were under 22 or over 50 years of age, with no difference in gender. The majority (33) of clients were 'repeat testers' who tested regularly, following the Tanzanian Ministry of Health guideline to test every three months when at risk and every six months to one year when not at risk. Re-testing is therefore considered normal practice under Treat All. Amongst the 33 'repeat testers', most had previously tested between 1 and 4 times. There were some exceptions: 3 repeat testers from the urban area had tested more than 10 times. Numbers were too small to indicate clear gender differences. Not presented in the table are some other characteristics of the participants of the rapid interviews. Most (30) of the CBTC clients we interviewed were married. Seventeen of the 41 participants tested for the first time through a community-based testing campaign, 21 had tested in a Health Facility for the first time, and 3 participants did not provide an answer. Amongst repeat testers, 14 had tested repeatedly at community-based testing campaigns; none could distinguish between CUAMM campaigns or campaigns organised by other organisations. Of the 8 HIV-negative clients who participated in a follow-up IDI, 5 were men, 4 of whom were over 50 years. Of the 3 women, 2 were in their mid-twenties.

Twenty-three IDI were conducted with 23 people who had tested HIV-positive in the outreach campaign. Of these 23, 10 (5 women, 5 men) were immediately linked to care following the Treat All strategy; 7 (6 women and 1 man) tested positive in the CBCT, but were not yet linked to care, or had linked to care but had dropped out. We additionally interviewed 6 people (5 women, 1 man), who re-tested in the outreach campaigns while already enrolled in an HIV treatment programme elsewhere (CUAMM testing registers showed that 13% of clients of CBTC were in fact already taking antiretroviral treatment). Table 2 presents the testing histories of the IDI participants who tested positive in the CBTC.

Amongst the 23 HIV-positive participants who tested positive in the CBTC, 10 were first-time testers, four had repeatedly tested before eventually testing positive; three already knew their status from a prior test but used the campaign to retest. Six positive clients were already enrolled in a treatment programme elsewhere. Of the 10 people who tested positive without having a prior awareness of their status through an earlier test, 5 were not yet linked to care or had initiated care but dropped out and were considering re-testing.

**Table 2. Testing histories of IDI participants who tested positive in the CBTC (n = 23).**

|  | Male | Female | Total |
|---|---|---|---|
| First-time testers in CBTC and being found HIV-positive | 4 | 6 | 10 |
| Repeat testers before being found HIV-positive | 1 | 3 | 4 |
| Repeat testers already aware of HIV-status but not linked to care | 1 | 2 | 3 |
| Repeat testers already on antiretroviral treatment | 1 | 5 | 6 |

## Outreach testing convenience

Testing campaigns were generally experienced by study participants as an easy way to access HIV testing, creating the convenience of closer-to-home testing when compared to traveling to several hours-away healthcare providers, or, in the urban site where services are closer, in comparison to having to arrange a clinic-visit. Outreach-facilitated HIV testing could easily fit in with farming or other day-work in town, or with social visits to family or friends and did not require planning or transport costs:

> [. . .] If it's here [testing in the community, there's] no need for preparations. If you have the intention, you just go as you are, you only [have to] wash your feet. But if you are going to the government hospital or dispensary you need to get highly prepared: bathing, taking tea, then to find transport and fare. All these things make a person feel lazy.
>
> — *Male, 71, HIV-negative, repeat tester, urban site*

Many repeat HIV testers were acquainted with regular testing and expressed inspiration to test again after hearing announcements in the village, through word of mouth, or because they were passing by the testing tents.

## Testing as a 'must': The routinization of testing

HIV-negative participants talked about testing in terms of 'controlling uncertainty'. Regular proof of negative status provided confidence, as one participant explained:

> I come from work and I test every now and then. I decided to pass by. . .The service gives you confidence in living without worries.
>
> — *Male, age unknown, married, sales supervisor, urban site, tested 12 times*

Repeat testers often framed testing as following advice from health services. This is likely a reflection of the fact that between 2004 and 2010 the national Angaza Zaidi campaign widely encouraged people who had tested HIV-negative to return after three months to confirm their negative status [12]

> We were informed by village leaders that health workers will come and test. When you are told to test every time, it is good to follow and respond. . .
>
> — *Female, 66, first-time tester, rural area*

In general, counsellors were following Tanzanian National AIDS Control guidelines and thus screened clients for potential risk behaviours. Based on such assessments, those getting tested were advised to return for a follow up test 1 month, 3 months, 6 months, or 1 year later. While interviewees said they were following advice, they also characterized regular testing as a 'need' or 'must'.

> I have a habit of testing regularly. I am told to retest every three months. I like the hospitality [of the testing campaign] and it was brought to my residence.
>
> — *Female, HIV-negative 24, student, unmarried, urban site, tested 4 times*

> I came to check that the virus truly is not in my body. Experts like you advise to test every three months. I tested 5 days ago at CUAMM; I was not told when to re-test again. I was

given a card but left it at home. When I find a place where people test, I must test, it does not matter when the last time I tested. I heard the music. I like that I can easily check my health status around here in the community.

— *Male, HIV-negative, 27, farmer, unmarried, rural site, tested 5 times*

Thus, HIV testing was framed as routine by counsellors in discussions with clients. Participants mentioned that testing was even a pre-requisite for membership in some organisations or for partaking in activities of that organisation. This was the case for a youth seminar held by a local organisation advocating entrepreneurial skills that often partnered with the implementing organisation:

A young HIV-positive woman came to the testing campaign in town because of an entrepreneur workshop. While in the workshop, it was announced that everyone should go for testing, after which all her companions queued. She was unable to discuss this with a counsellor before entering the tent because she did not want others to know her status. She . . . was scolded by the counsellor for wasting resources.

*Notes based on interview with 21-year-old woman, HIV-positive, tested while on ART, observation—urban site*

Religious institutions also stimulated testing, requiring it for marriage proceedings and for caretakers of young children. Despite these 'pressures to test', people were also routinely told that testing was an individual choice; interviewees emphasized that testing was their personal choice and that they had not been influenced by others.

## Testing because of feeling 'at risk'

Interviewees' knowledge about the risk of sexual partnerships was a major motivations to test. Many told stories of being placed in risky situations because of their partners; this was true of HIV-positive and HIV-negative people, as well as married and unmarried people. Men mentioned having regular sexual relationships with women other than their wives or mistrusting a partner as reasons to test. Most women said they did not trust their partners, or they wanted to know their health status for future prevention: 'to stay safe and protect myself'. Stories were often offered in explanation for repeat testing.

I mean, let me honestly tell you why I like to test. As I told you in the beginning, I am a woman who moves often. I can move from Shinyanga to Mwanza, then from Mwanza maybe I can move to Dodoma. I can meet with a man and have sex with him and he becomes my lover. You cannot know how that man lives, or whether he is safe or not because, young people of this generation, if you tell them: 'let's go and test', he tells you 'I am confident about myself'. So, you cannot know, you must have the courage to go to test.

— *Female, 26, unmarried, HIV-negative repeat tester*

A feeling of risk was also influenced by the behaviour, health, or HIV-status of a sexual partner. When a partner would refuse to go for testing, this raised suspicion and could motivate someone to get tested.

What pushed me most to go and test is because of the kind of husband that I am living with [. . .]. Because when I was telling him 'let's go for testing together', he'd refuse, saying he is

safe and, if it's about testing, I should go and test [alone]. So, then it was just my heart that kept on pushing me to go and test, just that.

— *Woman, 28, tested HIV-positive in campaign, linked to care*

In many of the interviews, testing was about monitoring, checking one's HIV status regularly. Public health messages, counsellors and healthcare providers promote regular testing, and interviewees knew that one 'good result' did not mean that one would stay HIV negative. While a negative status motivated people to 'stay safe' by emphasizing fidelity with their partners and using condoms with irregular partners, HIV-negative testers felt that 'being safe' was a temporary state. HIV-negative clients were told by counsellors that although they were 'safe today' they should continue to protect themselves.

Maybe previously the symptoms were not observable. You know the three months [between tests], you feel like, 'Maybe I was not yet. . .' Because they say: 'To be diagnosed it takes time'. So that's what makes me test every now and then, after being told after three months.

— *Male, 71, HIV-negative, repeated tester*

Feeling at risk was not just related to sexual partnerships. In the in-depth interviews, many noted they lived in an environment where sharp objects, bodily fluids through caring for HIV-positive relatives, or living together could put them at risk for infection. This was what motivated parents to test young children.

She [a 39-year-old woman bringing her children for testing] said 'If a child is cut with a sharp tool sometimes, they don't say at home. . . they might live with HIV infections without knowing, that is why I have brought them here to check their health status'. After the test results were negative the counsellor asked the parent to test them again in September next year.

*Observation report, rural testing campaign, 180926*

Shinyanga region has a highly mobile population, high rates of illiteracy, and high rates of early marriage. Because of these factors, counsellors advised those participating in testing campaigns to test every 3 months. However even clients who were not categorized as 'at risk' by counsellors, figured themselves at risk. Almost every family in Shinyanga had the experience of caring for a relative with HIV and observing the debilitating consequences of untreated HIV. In this general risk environment, having sexual relations and/or caring for those known to be HIV positive, but also everyday activities such as hand-washing clothes, which exposes people to the bodily fluids of people whose status is unknown, were mentioned as sources of worry. Repeated testing in this context was expressed as a way to maintain control in the face of uncertainty.

A 56-year-old man is a repeat tester. He enters the counselling tent and a conversation ensues in which it is clear that the man is not at much risk for infection: he reports to be married; he is 'done with extra relationships'. The counsellor suggests he come for testing once per year in the future and writes a test-date a year later, on the man's testing card. In the interview after this session the man is not happy. He states that he might run a risk for infection through engaging in farm work, going to hairdressers, and caring for HIV-positive people, and wants to check his health status every three months.

Interviewer: Don't you think that [..] you are not right, [that] it is like a waste of resources since you do not have any risk factors? [. . .]

Respondent: You may trust yourself that you are safe, but you never know what your partner has done, so of course that gives you worries. If you were alone then you could say: 'I am always here at my place', then ok.–*Male*, *71*, *HIV-negative*, *repeated tester*

## Testing 'to know early'

Direct experiences of having close family members who waited too long to get tested and died of HIV/AIDS shaped ideas of risk for HIV-negative repeat testers. Most had participated in prolonged caregiving tasks and this experience motivated regular testing in order to identify HIV positive cases early enough and prevent becoming physically dependent on others. While young people tested because they had parents who were HIV-positive or other relatives who had died, many older people had had direct caring experiences and tested because they did not want to burden their families.

When I saw my aunt was sick, that's when I got the confidence to go and test because that disease is very bad. [. . .]When she went to test she was severely sick. When she started to take treatment, I don't know, it's like the medicines did not work or what. . . It did not even take a month before she died.

— *Female 26, HIV-negative, repeated tester*

The thing that makes me test every now and then is to know what exactly my current status is. If I learn I am not okay, I will hurry up for treatment [. . .]. What brings those feelings is the experience of living in a family. . . like if you have a family of many people, especially adults who are sick, that will make you hurry up for early testing.

— *Male, 62, HIV-negative, repeated tester*

In counselling sessions, starting treatment early is framed as being able to stay healthy and live life as normal. In Shinyanga, community members had hands-on experience with the devastating consequences of HIV. Stigma around HIV remains high because of its association with inevitable death, and multiple sexual partnerships. Early diagnosis and early start of treatment then is also associated with the ability to keep one's HIV status a secret and to die with dignity. But while living and dying with dignity was an important motivation for HIV testing, the main reason people gave for repeat testing was that it allowed people to continue living with hope; catching the disease early meant being able to better treat it. A 71-year-old HIV-negative man from urban Shinyanga, who had witnessed the death of many people and was taking care of orphaned children at the time of research, associated hope with early treatment:

It's not like you wait until when a person is weaker and weaker, but as soon as you are diagnosed, treatment begins. It helps. Rather than staying with the hope that 'I am still healthy, no problem', until when you are too weak to start, waiting for CD4 to decrease and start treatment. . .you are already discouraged and approaching death.

— *Male, 71, HIV-negative, repeat tester*

This same man was one of very few people who discussed early treatment as a form of prevention: 'It helps a lot, yes and when they are using (treatment) it helps not to transmit to the other'.

Community leaders also describe a change in the way HIV-testing has come to be seen in the age of Treat All: it has become unfashionable to suffer.

> At the moment a person gets to a stage where he cannot get out of bed because of HIV/AIDS, people nowadays call it suffering the 'old-fashioned way'. And it is an embarrassment, because nowadays people do not suffer in such a way. They'd rather test for HIV, then use medication and have a proper death. [. . .] So lately people in the community feel bad leaving their families with so many sorrows, with such a bad type of death. They'd rather test for the virus to avoid shame and a horrible death'.

> Focus group with community leaders, Shinyanga urban site, 180529

Early treatment initiation is thus also framed as a moral duty to remain healthy for the sake of planning one's life and out of responsibility towards family and one's partner.

Counsellors emphasize this message, especially to those diagnosed with HIV. They are told that the diagnosis does not mean the end of life and that by starting treatment early they can live a long and healthy life. Amongst those who tested HIV-positive through the campaign, the desire to start treatment early was clouded by fear of what treatment might mean for their everyday well-being. Antiretroviral drugs were recognized as 'strong medicines' and many remembered people who died just after starting medication. While the UTT message is widely spread and accepted in theory, the reality of starting treatment early is complicated.

> When you use treatment early you keep your body well, nothing will happen to you. Because if you delay, and there are also problems when you go severely [become severely sick] [. . .], everyone will surely know, eh, she was this [HIV-positive]. You know us Tanzanians? [. . .]. But when I am okay [healthy], like the way I am?, who will notice me? Nobody will notice. . .. I will start [medication].

> — *Woman, 25, HIV-positive, not linked to treatment*

## Testing as 'proof'

A specific motivation for using the testing services at CBTCs was to prove knowledge about the virus' presence in the body to oneself or to one's partner. For young people, testing was related to the fear that parents might not tell youths if they had received a positive diagnosis. Testing was a means to gain knowledge.

> You might have been born with HIV, but you don't know your HIV status and maybe nobody is telling you. Therefore, it is good to test when those testing opportunities come around you in the community.

> — *Female, 16, first-time tester, rural site, observation 180905*

Some participants had already tested positive elsewhere but did not believe the result or needed time to come to terms with the results.

> Mmmh, let us say what influenced me: it's when I started living with a woman. The woman asked me to go for a test, then we decided to go and get tested. I was then found infected. After being diagnosed with the infection, honestly, I was not shocked or hurt so much. Instead I felt that maybe these people were deceiving me with their test kits, so we just ignored it. Two or three years later I tested for the second time to prove whether it was true

or not. [. . .] When I was looking at how healthy I was and what they were telling me, being infected, for sure I did not believe it. The type of jobs I do, it is tough work, but I am managing it without any problems or falling sick.

— *Male, 35, HIV-positive, linked to care, urban site*

People sometimes had trouble accepting an HIV-positive result when their bodily experience was of being healthy and strong, as many continued to associate HIV with a loss of strength and prolonged illness. Experiencing ill-health was a primary motivation for testing amongst clients that tested HIV-positive at the campaign, and again testing was used as a way to get proof.

What motivated me was that I used to fall sick most of the time at home, sometimes diarrhoea, headaches. My wife said: 'let's go for testing'. There were some service providers [community testing campaign], that's why we came.

— *Male, 45, HIV-positive, linked to care*

Some people retested, even though they were already enrolled in treatment. While one person tested to 'prove' to her disbelieving new partner that she was indeed HIV positive, tests were most often used to prove how the body was progressing on treatment, perhaps confusing the HIV test with viral load testing, or presuming that treatment could eventually result in a negative test outcome.

I was sick, my body was not in its normal condition. [. . .] I said to myself that all people are going for testing, even those who have never tested before. [. . .] They also took me for testing. So, I decided to test too. [. . .] [The counsellor] told me that 'you have the HIV virus'. [. . .] I wanted to know if the viruses are decreasing.

— *Female, 70, HIV-positive, tested while on treatment*

A final way that testing was used as proof was to confirm or deny HIV status in new relationships or as part of marriage proceedings.

What made me go there was because I had a husband who died. I have been all alone and thought that I should get a partner to help me. So as days went on I got someone. [. . .] He was a pastor and even his fellow pastors told him to first get tested before proceeding with anything else. [. . .] He told me to wait until he comes [to Shinyanga], we would go for testing. But before he would come, I thought to myself that I would be ashamed if he would use transport fare, so I'd rather go for testing by myself first. So I went and tested and was found with that condition. I was infected.

— *Female, 58, HIV-positive, Linked*

## Discussion

The shift to Treat All as the standard of care in Tanzania in 2016 formed an important background to this study, because it encompassed a shift in practice: all people found to be HIV positive were to be directly initiated onto treatment, regardless of their health status. As part of Treat All, the government also rolled out the promotion of testing through the 2018 nationwide 'Furaha Yangu' campaign [13], which aimed at increasing testing especially amongst men and youth.

Within this broader context, this qualitative ethnographic study investigated the motivations of both HIV-positive and HIV-negative community members to test repeatedly through community-based outreach campaigns. We found that repeat testing for both groups was about proof: results could provide information about relationships, bodily sensations, and the workings of ART.

Amongst people who tested positive, our findings confirm those of previous studies amongst people living with HIV: repeat testing diminishes doubt about an earlier diagnosis or confirms suspicion about the cause of ill-health [14]. Clients who test positive while already on ART re-tested to settle doubts about their diagnosis or to see if the public health message that 'the virus goes to sleep' when on treatment was true. Our findings suggest that, in order to stimulate engagement with HIV services, counselling for people who test HIV-positive must address both doubt and the workings of ART explicitly, while also considering that accepting a positive diagnosis may take time.

For HIV-negative repeat testers, outreach campaigns, with their easy access, provided the means to address 'doubts' created by 'risky' situations, which spurred preventive behaviour and more repeat testing [15]. People's understandings of 'risk' differed from a public health understanding of risk as sexual transmission of HIV. For repeat testers, risk was related to uncertainty about partners in the context of high mobility, the presence of bodily fluids on everyday objects like clothes and sharps, and a history of witnessing suffering. For them, receiving an HIV-negative result provided temporary re-assurance. The message to screen regularly for HIV had been ingrained in the minds of many who participated in testing campaigns, and repeat testing was understood as following government advice. Counsellors and community leaders tasked with mobilization framed regular testing as a moral responsibility for health and family well-being: knowing early allowed one to preserve health, to not become a burden to one's family, and to die with dignity.

Susan Reynolds Whyte has described rapid diagnostic devices such as HIV tests as having 'social lives'; diagnostic devices are used for more than just diagnosis, they are used for different kinds of surveillance of bodies. Whyte and colleagues argue that the increasing use of diagnostic devices, tests to diagnose HIV and determine CD4-count, has become incorporated in people's understandings of the body (Whyte, 15-02-2017 Public lecture). The findings from this study build on their analysis. The 'proof' provided by tests is a form of self-surveillance of the bodies of HIV-negative and HIV-positive people, self-screening that is embedded in social dynamics and responsibilities, but it is also used as a kind of social currency in relationships. Participants of the testing campaigns 'transform' the public health goal of testing from case-finding and prevention, to self-screening and surveillance. In the evaluation of a public health intervention, such as community outreach testing, and whether it is effective, it is worth considering the parameters of 'effectiveness'. As Hardon and Dilger argue, AIDS treatment services are always transformed by users and local implementers to fit local needs [16].

The current shift away from general population testing, towards targeted testing of at-risk groups [17] and people in high-risk locales, makes sense if effectiveness is measured in terms of HIV-positive people found. Recent evidence from three African UTT trial sites show that contrary to expectation, general population testing has a limited effect on population-level HIV incidence [17]. Targeted approaches are therefore considered more cost-effective than general population testing, which is important considering the dwindling global funding for HIV treatment programmes [18].

While repeated testing may not seem cost-effective in the context of high ART coverage nor yield many new HIV cases, the rationale of HIV-negative repeat testers highlight important outcomes of testing campaigns beyond detecting new HIV cases. They show a wish for regular health monitoring, which is crucial for prevention [19].

UTT trials to date have found that proximity to services is a crucial factor in people's decisions to test repeatedly for HIV. Nearby services attract people with fewer economic resources [20] as well as those with increased risk exposure [21]. Before discouraging repeat testing out of concerns for cost-effectiveness or epidemiological reasoning, it is crucial to understand how community members engage with community-based testing campaigns, so that they may be tailored into a service that meets health-related needs that exceed HIV detection.

At the global level, there is a growing recognition that what is needed to identify the remainder of HIV cases is a community-based approach, where HIV testing and care services are tailored to the needs of, and identified by, specific groups of community members, and where HIV services may be integrated into broader health care services [22]. There is a growing consensus that while the uniqueness of the HIV response in including the right to health of often highly marginalized groups should be safeguarded [23], HIV services should increasingly be incorporated into UHC models. Routine health screening using such HIV outreach models offers great potential for the community management of hypertension and diabetes and other non-communicable diseases (NCDs), which are on the rise in Tanzania and for which the primary care system is ill-prepared [24]. Studies in both Tanzania and Uganda have shown the successful integration of HIV and NCDs screening [25]. While the discussion of UHC is currently dominated by financing considerations, the role of community engagement with health services and increasing trust between health service providers and communities is equally important [26, 27].

We find ourselves on the 'eve' of a shift in public health thinking about HIV, moving away from general population to more targeted approaches of individuals considered to be at risk. While this shift is underway, it is worth taking a step back and thinking about the potentially 'normalizing' and destigmatizing effects of population-wide screening for health beyond HIV. In other words, general population outreach programs originally started for HIV could be expanded to include more (wellness) services, including determination of risk factors for NCDs, like elevated blood sugar and blood pressure. This will allow for more timely diagnosis of patients who are chronically ill and prevent unnecessary UHC expenditures on NCDs when diagnosed (too) late. Moreover, there is a social price for targeted HIV testing (like index-testing), which may stigmatize members of epidemiologically defined risk groups, resulting in their 'hiding' from the formal healthcare system. There are ample examples of UTT approaches failing to reduce HIV incidence in Rwanda, Ethiopia, Botswana [17]. In this light and with our current data on community motivation to participate in HIV testing, we question whether the current phasing out of general population testing is wise considering the move towards UHC?

## Conclusion

People who participate in HIV testing campaigns in remote Tanzania do so to monitor their health to achieve (temporary) reassurance, to alleviate feelings of being at risk due to suspicions about their partners, and out of feelings of responsibility. Tests are used as proof: to confirm a suspicion of ill health, to confirm an earlier diagnosis, and to gain general knowledge about the body. The implementation of community-based testing campaigns in Shinyanga and Simiyu Regions shows how an originally top-down public health approach has been repurposed by those who participated. Half of the clients reached by the campaigns were repeat testers, and the majority were HIV-negative. Repeat testers have internalized the general public health message that regular testing and knowing of one's health status is beneficial. Testing in the era of Treat All is also used to 'know early' and is thus framed in terms of social dignity and moral responsibility towards others. While community-based testing does not result

in identifying a sufficiently high number of new HIV cases, it does function as an important tool in routine health monitoring, one with great potential for integrating routine health screening on the way to UHC.

## Supporting information

**S1 File.**
(ZIP)

## Acknowledgments

We thank all our institutional collaborators, the study participants, the staff at the project clinical sites and laboratories, as well as the project support staff for their invaluable support to this program in general and the current manuscript in particular. The content of this manuscript is solely the responsibility of the authors and does not necessarily represent the official views of any of our implementing partners.

## Author Contributions

**Conceptualization:** Josien de Klerk, Eileen Moyer.

**Data curation:** Judith Meta, Tusajigwe Erio.

**Formal analysis:** Josien de Klerk, Judith Meta, Tusajigwe Erio, Eileen Moyer.

**Funding acquisition:** Tobias Rinke de Wit.

**Investigation:** Judith Meta.

**Methodology:** Josien de Klerk, Judith Meta, Eileen Moyer.

**Project administration:** Josien de Klerk, Tobias Rinke de Wit.

**Supervision:** Josien de Klerk, Tobias Rinke de Wit, Eileen Moyer.

**Validation:** Arianna Bortolani.

**Writing – original draft:** Josien de Klerk, Tusajigwe Erio, Eileen Moyer.

**Writing – review & editing:** Josien de Klerk, Arianna Bortolani, Tusajigwe Erio, Tobias Rinke de Wit, Eileen Moyer.

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
