## [Decision Letter · Decision Letter 0]

26 Apr 2021

PONE-D-21-07606

‘It is not fashionable to suffer nowadays’: Community motivations to repeatedly participate in outreach HIV testing indicate UHC potential in Tanzania

PLOS ONE

Dear Dr. Moyer,

Thank you for submitting your manuscript to PLOS ONE. After careful consideration, we feel that it has merit but does not fully meet PLOS ONE’s publication criteria as it currently stands. Therefore, we invite you to submit a revised version of the manuscript that addresses the points raised during the review process.

We look forward to receiving your revised manuscript.

Kind regards,

Bidhubhusan Mahapatra, Ph.D.

Academic Editor

PLOS ONE

Additional Editor Comments:

Very informative and well written paper. I have only two suggestions. First, some of the quote texts are too lengthy. I suggest authors to reduce the length wherever possible. Second, I suggest authors to include a conceptual framework based on the study findings, if possible.

Journal Requirements:

2. For qualitative studies, PLOS ONE suggests  consulting the COREQ guidelines: http://intqhc.oxfordjournals.org/content/19/6/349 to ensure that all relevant information is provided . In this case we would appreciate more information about: the number  and training of interviewers; how participants were selected ( we note that also community leaders quotes are reported; please describe in the methods how they were selected and recruited);  if a pilot study was tested; how data was coded; if bias issues were considered. Moreover, please provide the interview guide used (in the original language and in English) as a Supplementary file.

3. Please ensure you have included the registration number for the clinical trial referenced in the manuscript.

4. Thank you for stating the following in the Financial Disclosure section:

"The Shinyanga and Simiyu Test & Treat program in Tanzania is supported by Gilead Sciences (USA) https://www.gilead.com/purpose/giving/funding-requests

and the Diocese of Shinyanga through the Good Samaritan Foundation (Vatican). https://www.humandevelopment.va/en/il-dicastero/fondazioni/il-buon-samaritano.html

Grant Number: xxxxx

TRdW is a co-recipient of this this award. All other authors were employed under its auspices. The funders did not play an active role in study design, data collection, analysis, decision to publish or preparation of the manuscript."

We note that you received funding from a commercial source: Gilead Sciences.

Reviewers' comments:

Reviewer's Responses to Questions

**Comments to the Author**

1. Is the manuscript technically sound, and do the data support the conclusions?

Reviewer #1: Yes

Reviewer #2: Yes

2. Has the statistical analysis been performed appropriately and rigorously? 

Reviewer #1: N/A

Reviewer #2: N/A

3. Have the authors made all data underlying the findings in their manuscript fully available?

Reviewer #1: Yes

Reviewer #2: No

4. Is the manuscript presented in an intelligible fashion and written in standard English?

Reviewer #1: Yes

Reviewer #2: Yes

5. Review Comments to the Author

Reviewer #1: Reviewer’s comments

1. The study presents the results of the original research.

Reviewer: Yes

2. Results reported have not been published elsewhere.

Reviewer: Yes

3. Experiments, statistics, and other analyses are performed to a high technical standard and are described in sufficient detail.

Reviewer: Yes

4. Conclusions are presented appropriately and are supported by the data.

Reviewer: Yes

5. The article is presented in an intelligible fashion and is written in standard English.

Reviewer: Yes

6. The research meets all applicable standards for the ethics of experimentation and research integrity.

Reviewer: Yes

7. The article adheres to appropriate reporting guidelines and community standards for data availability.

Reviewer: Yes

Additional comments.

Introduction: The citation on page # 3, line 65 makes it difficult for the reader to follow. I suggest that it should be inserted at the end of the sentence (e.g., urban Shinyanga (reference).

Re-cast the citations on page # 3, line 69-70 (e.g., Sharma et al. 2015; Sabapathy et al. 2012)

Define “ yields” on page # 4, line 78.

A citation at the end of the sentence “….Tanzania is in transition period…..” is missing, on page # 4, line 82-83.

Move the 51.1% just after the words “…At least half (51.1%) of the population…..”, On page # 4, line 85.

Define “ethnographic study” on page # 4,line 92.

It will be beneficial for the readers to know how information from this study will be utilized in the study setting (i.e., who are the primary, secondary, and tertiary beneficiaries). I suggest this should be described after the last sentence on page #4, line 94.

Study setting and project background: On page # 5, line 98, kindly provide the approximated population of the Shinyanga Region, and the male: female ratio.

On page # 5, line 102-103, kindly provide (n=) for government dispensaries, Heath clinics, private faith-based Heath centres, and private laboratories. And add a citation to support that data.

On page # 5, line 111-the authors provide the distance-time by cycling….I was curious why only cycling, and not walking/or by a motor vehicle? Any justification for that?

On page #5, line 116-the authors mentioned in town, work and school hours affect….I suggest that providing the time range(e.g, from 7.00 am-4.30 pm) will make it clear for a reader to follow.

Data collection: On page #6, line 123, the authors should provide the number of trained Tanzanian researchers who conducted the structured observations (e.g., N=? male: female ratio).

On page # 6, line 140-the authors should provide examples of questions asked for each category (e.g., their testing experiences, their motivation for testing…etc.).

On page # 6, line 143-142, the authors mentioned that participants were asked about the new Treat All policy. How did participant get access to the new policy?

Did the data collection tools pilot test before data collection? I suggest the authors should provide this information and provide the number of participants who participated in the pilot testing and how was the information from the pilot test used.

Data analysis: On page #6, line 145, the authors used Qualitative content analysis. I suggest the authors should justify using content analysis instead of other alternative data analysis methods.

On page #6, lines 148-149, the authors describe how the analysis was done. I would like the authors to provide more information for the following: (1) What were the major coding categories?, (2 ). Did the study assessed coder consistency between coders?, (3). Post-coding: did coders examined, compared, and contrast the distribution of themes within and across subgroups, and whether this is reflected in the results section?, (4). Did the in-depth interviews transcripts and direct observation notes analyzed together or separate and later combined?. I suggest authors should describe how they handled this. (5). I did not see mentioned how the study observed reflexivity (i.e., conscious self-reflection to make explicit individual's potential influence on the research process), and steps taken to ensure study rigour. I suggest the author address these two important issues in the qualitative study.

Results: A table of participant characteristics would be beneficial to the readers.

Reviewer #2: This is a very well written manuscript. I really like the title which attracted me to chosse to review this manuscript. The issue is important in the context of test and treat and overobsession with yield. I really like the discussion session and the recommendation to use the HIV outreach testing spaces to integate other diseases in th era of UHC.

In addition, the autors can make few minor changes

1. Add one or two Tanzanian co authors. As the study was conducted and implemented in Tanzania, I am sure there may have been Tanzanian researchers who have either contributed or can contribute to this paper

2. It would be good to provide a bit more information about the context. The risk behaviours of people in the region, vulnerability factors and AIDS related deaths. AIDS related deaths are mentioned later but as I was reading, I felt that I needed to understand the context a bit more which would help me understand the respondents testing behaviours and motivations

3. In the method section, I got a bit confused about the different methods. It may be good to either present the method and the sample in a tabular form or a bullet form for the reader to understand

4. Though there were quotes of men and women across age, I was looking for some differences based on gender in the results and discussion section. The paper seemed blind to differences in motivation based on gender though literature does indicate that motivation of testing among men and women are different. A bit more exploration using a gender lens may improve the paper

5. In the discusison section while I like the critique of doing testing to find PLHIV and link them to treatment, I was expectecting a bit more about the value of testing for prevention. That section can be expanded.

6. PLOS authors have the option to publish the peer review history of their article (what does this mean?). If published, this will include your full peer review and any attached files.

Reviewer #1: No

Reviewer #2: No

---

## [Author Response · Author response to Decision Letter 0]

3 Nov 2021

We've included our detailed response to the reviewers' previous comments below. Here, we reply to the most recent request from the editors. Namely, we have amended the list of authors as requested, added the ethics statement to the Methods section of the manuscript, and adjusted the data availability statement to indicate that the secretariaat (not t.rinkedewit) should be contacted to access data. 

Thank you for your detailed comments on the manuscript PONE-D-21-07606

‘It is not fashionable to suffer nowadays’: Community motivations to repeatedly participate in outreach HIV testing indicate UHC potential in Tanzania, 

We feel the comments have improved the solidity of the manuscript. Below we have provided detailed questions to all concerns raised. 

1. Additional Editor comments:

Very informative and well written paper. I have only two suggestions. First, some of the quote texts are too lengthy. I suggest authors to reduce the length wherever possible. Second, I suggest authors to include a conceptual framework based on the study findings, if possible.

 We have adjusted the lengths of quotes where possible. We found it not possible to include a conceptual framework. 

2. Response to Journal Requirements:

We have checked the reference list. We added a reference of the Tanzanian household census following a comment about the missing population data

We also added a reference of the National Guidelines for the Management of HIV and AIDS sixth edition, October 2017 following the comment about the missing citation on Tanzania is transitioning to targeted HIV testing 

Three references did not meet the criteria for inclusion in the reference list. We address these in a separate letter entitled: letter about personal communication.

We have made sure the manuscript meets the style requirements

2. For qualitative studies, PLOS ONE suggests consulting the COREQ guidelines: http://intqhc.oxfordjournals.org/content/19/6/349 to ensure that all relevant information is provided . 

In this case we would appreciate more information about: 

- the number and training of interviewers;

- how participants were selected ( we note that also community leaders quotes are reported; please describe in the methods how they were selected and recruited); 

- if a pilot study was tested; how data was coded; 

- if bias issues were considered. 

- Moreover, please provide the interview guide used (in the original language and in English) as a Supplementary file.

Please find below a short answer to the raised points. We have adapted the method section accordingly

Number and training of interviewers

Two local social scientists, a man and a woman with an MA degree in sociology were selected and conducted all the interviewing. One of the social scientists was a native Kisukuma. They were trained to do structured observations and were provided a guideline. All interview guides were tested and translated into Kiswahili. The social scientists were trained in note-taking, probing, transcription and data-analysis in two data analysis workshops. 

Participant selection

We adapted the method section to include this information. Participants were selected through a two-step sampling framework. HIV-negative participants were selected during community-based testing campaigns in both a rural and in an urban location. The social scientists conducted rapid interviews with attendees and asked for consent to recontact the participants. A sample of repeat testers were then contacted by phone for a longer interview.

HIV-positive participants were contacted through counselors of the CBTC using lists of men and women who had tested positive in a community-based testing campaign. Here the counselors selected a sample of men and women from both a rural and an urban location. 

Community leaders from the communities surrounding both clinics (the urban and the rural clinic) were visited by the social scientists and asked to participate in a focus group discussion. 

Pilot study?

No pilot study was conducted. The tools were tested

How data was coded? 

Data was coded in two separate coding clinics of one week, one after three months of data collection: (observations during testing campaigns & rapid interviews) to decide on themes to explore in in-depth interviews 

Each interview was transcribed in the original language by a trained data transcriber, and checked against the tape by the local social scientists. The interviews were translated from Kiswahili to English by a translator. The entire team analysed the Kiswhahili version. 

A team consisting of the lead social scientist and the two local social scientists each read several transcripts and coded individually. We then compared the themes and developed a code book. Each code was discussed and defined in the NVIVO code book. We then applied the code book to a next set of interviews together and refined it. We then coded the remaining interviews individually.

Bias issues 

The following potential biases were acknowledged and are discussed in the revised text:

- Age and gender of the respondents for the IDI with HIV-negative testers. Recontacting proved difficult. 

- Rather than selecting equal numbers of male: female or youth – middle age – older, we sampled on categories of testing frequency, linkage to care history, lost-to-follow up/lost to linkage. We recruited as much as possible equal numbers of male and female but were dependent on who granted us permission. Because we wanted to select only people tested in a community-based testing campaign some of whom were not enrolled in care, recruitment was a challenge. How these biases affected the data was considered 

Moreover, please provide the interview guide used (in the original language and in English) as a Supplementary file.

Please see attached supplementary files. 

3. Please ensure you have included the registration number for the clinical trial referenced in the manuscript.

The study was not a clinical trial but an intervention study. We have removed this language. It is not registered as clinical trial. It has received ethical clearance through the National Institute for Medical Research. The ethical clearance numbers for the years in which the study ran include: NIMR/HQ/R. 8a/Vol. IXI 2711; NIMR/HQ/R. 8c/ Vol. I/ 1207; NIMR/HQ/R. 8c/ Vol. I/ 1447

We address point 4 (competing interests) and 5 (data availability) in the cover letter. 

5. Author’s Responses to Reviewer Comments: 

Comments to the Author

1. Is the manuscript technically sound, and do the data support the conclusions?

Reviewer #1: Yes

Reviewer #2: Yes

2. Has the statistical analysis been performed appropriately and rigorously?

Reviewer #1: N/A

Reviewer #2: N/A

3. Have the authors made all data underlying the findings in their manuscript fully available?

Reviewer #1: Yes

Reviewer #2: No

4. Is the manuscript presented in an intelligible fashion and written in standard English?

Reviewer #1: Yes

Reviewer #2: Yes

5. Response to Review Comments to the Author

Reviewer’s comments

1. The study presents the results of the original research. 

Reviewer: Yes

1. Results reported have not been published elsewhere. 

Reviewer: Yes

1. Experiments, statistics, and other analyses are performed to a high technical standard and are described in sufficient detail. 

Reviewer: Yes

1. Conclusions are presented appropriately and are supported by the data.

Reviewer: Yes

1. The article is presented in an intelligible fashion and is written in standard English.

Reviewer: Yes

1. The research meets all applicable standards for the ethics of experimentation and research integrity.

Reviewer: Yes

1. The article adheres to appropriate reporting guidelines and community standards for data availability.

Reviewer: Yes

Additional comments.

Introduction: The citation on page # 3, line 65 makes it difficult for the reader to follow. I suggest that it should be inserted at the end of the sentence (e.g., urban Shinyanga (reference).

Good suggestion, we changed it

Re-cast the citations on page # 3, line 69-70 (e.g., Sharma et al. 2015; Sabapathy et al. 2012)

Thank you, we changed this

Define “ yields” on page # 4, line 78.

Done

A citation at the end of the sentence “….Tanzania is in transition period…..” is missing, on page # 4, line 82-83.

we revised the sentence so it is more clear what we mean. 

Move the 51.1% just after the words “…At least half (51.1%) of the population…..”, On page # 4, line 85.

We moved this

Define “ethnographic study” on page # 4,line 92. 

Defined

It will be beneficial for the readers to know how information from this study will be utilized in the study setting (i.e., who are the primary, secondary, and tertiary beneficiaries). I suggest this should be described after the last sentence on page #4, line 94.

We included this information in the place you suggested:

Every three months the local social scientists attended the meetings of the community based testing teams of the local organization CUAMM. During these meetings they presented the focus of their work so far and the themes emerging from the results. Thereafter followed a discussion on these themes with the counselors and their supervisors. 

The interim findings and progress of the study were also reported in the quarterly meeting held at the offices of the regional medical officer where representatives from the local clinics and other district authorities were present. 

Beneficiaries:

- Local counselors of the testing teams of the implementing organisation

- HIV-partners/ government authorities in the district and region

Study setting and project background: On page # 5, line 98, kindly provide the approximated population of the Shinyanga Region, and the male: female ratio.

We added this information from the final census, conducted in 2012

On page # 5, line 102-103, kindly provide (n=) for government dispensaries, Heath clinics, private faith-based Heath centres, and private laboratories. And add a citation to support that data.

We obtained this information from the Regional AIDS Control Office and included it. Published figures only exist for Tanzania as a whole

On page # 5, line 111-the authors provide the distance-time by cycling….I was curious why only cycling, and not walking/or by a motor vehicle? Any justification for that?

Local clinics offered services to people coming from very remote places. Most people used bicycles to come to the hospital. In the clinic the counselor measured distance in terms of cycling. We added this

On page #5, line 116-the authors mentioned in town, work and school hours affect….I suggest that providing the time range(e.g, from 7.00 am-4.30 pm) will make it clear for a reader to follow.

School and work hours varied but the testing campaign programme planned the testing events both during the day as well as in the evening and on public holidays to also cater to people who could not attend day time events. We explained this.

Data collection: On page #6, line 123, the authors should provide the number of trained Tanzanian researchers who conducted the structured observations (e.g., N=? male: female ratio).

Two trained Tanzanian social scientists, a male and a female conducted the structured observations. We rewrote the data section for more clarity.

On page # 6, line 140-the authors should provide examples of questions asked for each category (e.g., their testing experiences, their motivation for testing…etc.).

We included some questions in the text: 

Questions per category included:

Testing experiences & motivation: 

1. Can you tell me about the day you tested, what convinced you to go to get tested? How did the counselor do the test? (probe for the testing practice)

2. How did you feel about this?

3. Was there anybody that helped you make that decision.

4. What did the counselor tell you? (probe about life, staying healthy, starting treatment)

Testing under Treat All

1. What do you know about Test And Treat? (probe for health and prevention)

2. If you compare to the past, before the Test and Treat started, what are differences in how people go for testing?

3. Have your ideas about testing changed compared to before?

On page # 6, line 143-142, the authors mentioned that participants were asked about the new Treat All policy. How did participant get access to the new policy?

In 2017 the Tanzanian government switched to a Treat All policy which was communicated through community leaders, health providers and through public messaging. By 2018 most people had heard in some way of the policy. Most people received the treat all message also from the counselor who tested them. The policy we speak about is the national policy. The project brought testing to remote places. We addressed this in the introduction

Did the data collection tools pilot test before data collection? I suggest the authors should provide this information and provide the number of participants who participated in the pilot testing and how was the information from the pilot test used.

We pretested the tools. For the rapid interviews we conducted three rapid interviews during testing campaigns and adapted questions that were unclear. The in-depth interviews were also pre-tested to check for flow and clarity of the questions. Because it was very difficult to find participants the pre-test was only done with a limited number of people. 

Data analysis: On page #6, line 145, the authors used Qualitative content analysis. I suggest the authors should justify using content analysis instead of other alternative data analysis methods

We explained more concretely what we did in this section. 

On page #6, lines 148-149, the authors describe how the analysis was done. I would like the authors to provide more information for the following: (1) What were the major coding categories?, (2 ). Did the study assessed coder consistency between coders?, (3). Post-coding: did coders examined, compared, and contrast the distribution of themes within and across subgroups, and whether this is reflected in the results section?, (4). Did the in-depth interviews transcripts and direct observation notes analyzed together or separate and later combined?. I suggest authors should describe how they handled this. (5). I did not see mentioned how the study observed reflexivity (i.e., conscious self-reflection to make explicit individual's potential influence on the research process), and steps taken to ensure study rigour. I suggest the author address these two important issues in the qualitative study.

We addressed these comments together with the additional reviewers comments and rewrote the data analysis section to address these valid questions

Results: A table of participant characteristics would be beneficial to the readers.

We provided two tables, one for the participants of the rapid interviews and one for the HIV-positive clients who were interviewed through an IDI. We focused on testing characteristics in the tables and provide more detail regarding gender and marital status in the text. We also rewrote this section for clarity.

---

## [Editor Report · Decision Letter 1]

2 Dec 2021

‘ It is not fashionable to suffer nowadays’: Community motivations to repeatedly participate in outreach HIV testing indicate UHC potential in Tanzania

PONE-D-21-07606R1

Dear Dr. Moyer,

We’re pleased to inform you that your manuscript has been judged scientifically suitable for publication and will be formally accepted for publication once it meets all outstanding technical requirements.

Kind regards,

Bidhubhusan Mahapatra, Ph.D.

Academic Editor

PLOS ONE
---

## [Editor Report · Acceptance letter]

6 Dec 2021

PONE-D-21-07606R1 

 ‘It is not fashionable to suffer nowadays’: Community motivations to repeatedly participate in outreach HIV testing indicate UHC potential in Tanzania 

Dear Dr. Moyer:

I'm pleased to inform you that your manuscript has been deemed suitable for publication in PLOS ONE. Congratulations! Your manuscript is now with our production department. 

Kind regards, 

on behalf of

Dr. Bidhubhusan Mahapatra 

Academic Editor

PLOS ONE